# Effects of an Electric Field on the Conformational Transition of the Protein: Pulsed and Oscillating Electric Fields with Different Frequencies

**DOI:** 10.3390/polym14010123

**Published:** 2021-12-30

**Authors:** Qun Zhang, Dongqing Shao, Peng Xu, Zhouting Jiang

**Affiliations:** Department of Applied Physics, China Jiliang University, No. 258 Xueyuan Street, Xiasha Higher Education Zone, Hangzhou 310018, China; zhangqun19970217@163.com (Q.Z.); dongqing1756@163.com (D.S.); xupeng@cjlu.edu.cn (P.X.)

**Keywords:** molecular dynamics simulation, electric field, conformational transition

## Abstract

The effect of pulsed and oscillating electric fields with different frequencies on the conformational properties of all-α proteins was investigated by molecular dynamics simulations. The root mean square deviation, the root mean square fluctuation, the dipole moment distribution, and the secondary structure analysis were used to assess the protein samples’ structural characteristics. In the simulation, we found that the higher frequency of the electric field influences the rapid response to the secondary structural transitions. However, the conformational changes measured by RMSD are diminished by applying the electrical field with a higher frequency. During the dipole moment analysis, we found that the magnitude and frequency of the dipole moment was directly related to the strength and frequency of the external electric field. In terms of the type of electric fields, we found that the average values of RMSD and RMSF of whole molecular protein are larger when the protein is exposed in the pulsed electric field. Concerning the typical sample 1BBL, the secondary structure analysis showed that two alpha-helix segments both transit to turns or random coils almost simultaneously when it is exposed in a pulsed electric field. Meanwhile, two segments present the different characteristic times when the transition occurs in the condition of an oscillating electric field. This study also demonstrated that the protein with fewer charged residues or more residues in forming α-helical structures display the higher conformational stability. These conclusions, achieved using MD simulations, provide a theoretical understanding of the effect of the frequency and expression form of external electric fields on the conformational changes of the all-α proteins with charged residues and the guidance for anticipative applications.

## 1. Introduction

With the rapid development of science and technology, the widespread applications of electronic devices play an important part of our everyday lives. The investigation of the effects of radio frequency and microwave electric/electromagnetic radiation fields on biological tissues has become a very interesting research topic [1,2,3,4,5]. One of the most crucial and still unclear problems in biophysics is how a protein folds into its native three-dimensional structure to exert biological function. The molecular architecture ensures unique spatial arrangements of charged, polar, and hydrophobic amino acids controlling structure and dynamics of the biopolymer and defining specific interaction domains with possible reaction partners by non-covalent intra-molecular interactions [6]. Due to any intrinsic or extrinsic factors, electrostatic interactions play a prominent role, although its impact on protein structure and function are poorly understood [7,8,9]. Such changes may occur, for instance, upon light absorption of protein-bound cofactors leading to intramolecular charge separation or isomerization of the chromophore that causes a reorientation of a charged or polar group within the protein matrix. Many of these local changes of the electrostatics may affect protein properties such as structures, redox potentials, and p*K_A_* values, even quite remote from the origin of the perturbation. Beyond the huge amount of studies on protein structure and stability, the inherent mechanisms and their relationships to experimental observables are still a challenge for the scientific community [10,11]. Recently, both theoretical and experimental studies have confirmed that, in addition to the thermal effects, macromolecules undergo electric field induced non-thermal conformational changes [1,2,12]. Such fields can affect the structural and functional stability of a protein by altering its conformation and can induce reversible changes in its activity [13]. Among food-processing techniques, external stresses are applied using thermal and non-thermal processes, including pulsed electric field and high electric field processing methods [14,15]. These processes help increase the shelf life of food products and improve their organoleptic properties by producing conformational changes in the protein structure [16]. Therefore, the research on the effect of electric fields applied to proteins is not only important as a scientific issue but is also very meaningful for industrial applications, such as food processing, protein designing, biological function controlling, etc.

The protein response to an external field is involved in protein folding/unfolding, protein aggregation, protein adsorption, protein recognition process, etc. [17,18,19,20,21]. Experimental studies were carried out to investigate the effect of an external environment on the protein properties and even to explore possibilities to control protein process via surface-enhanced resonance Raman spectroscopy (SERRS) [22], quartz crystal vibrational analysis (QCV) [23], measurement of electrical double layer capacitance [24], etc. Although these methods could partly give specific information about protein behaviors under an applied potential, difficulties are usually encountered in simultaneously applying electric fields and measuring protein behavior due to the limitation of measurement techniques [19,25]. Some experimental results are not consistent because of the intrinsic difficulties of ensuring homogeneous field application. Therefore, the research on proteins exposed in electric fields is ongoing and some phenomena are still not well understood.

In addition to existing experimental studies on electric effects of protein denaturation and stability, computational biophysics has started to focus on researching the inherent mechanisms of bio-molecules at atomistic level that cannot easily be observed from experiments [26]. Recently, meaningful research was carried out on the thermodynamics and dynamics of the protein by internal and external factors, such as properties of protein, surface and solvent, confinement, and electric fields using Molecular Dynamics (MD) or Monte Carlo (MC) simulations [27,28,29,30,31,32,33,34,35]. Efforts were made to understand the effect of external-field exposure on various kinds of proteins by means of the rapid development of computational techniques. Studies by Wang et al. showed that the secondary structure of insulin was disrupted under the application of an electrical field with strength higher than 0.25 V/nm [28]. The MD simulations were also performed to investigate the effects of external pulsed and static electric fields on protein folding and unfolding in Myoglobin [29]. The unusual stability of the soybean trypsin inhibitor protein in the oscillating electric fields was also evaluated by molecular modeling [34]. A typical beta-hairpain peptide chignolin exposed in the external static electric field and oscillating field was investigated by Astrakas et al. [36,37]. Although computer simulation becomes one of the powerful methods to discover inherent biological molecular mechanisms, some results are not consistent with the experiment. Moreover, previous research mainly focused on the proteins classified as α + β and α/β types with both α-helix and β-sheet on its secondary structural level. In order to contribute to comprehending the conformational transition, especially the evolutions of α-helical structures of the protein, we selected several all-α proteins as simulation samples. Besides the strength of the electric field, other characteristics, such as frequency and electric waveform, are seldom compared to the analysis of the effect of an external electric field on a protein. In this study, we used MD simulation to explore the effect of alternating electric fields, including pulsed and oscillating forms with different frequencies, on the conformational properties of all-α proteins. This work is organized as follows: following the introduction in Section 1, the methodology, including the construction of the simulation box and molecular dynamics simulation details are presented in Section 2. The simulation results regarding the effects of the frequency and the type of electric fields are discussed in Section 3. Section 4 contains the concluding remarks of this study.

## 2. Simulation Details and Methods

The initial configuration of object proteins (PDB code: 5CYT, 1BBL, 1FCS, 1F63 and 1EA8) were obtained from the Protein Database Bank. The detailed information of all protein samples used in this present article are given in Table 1. According to the Structural Classification of Proteins (SCOP) database, these proteins were classified as all-α proteins composed of 37 to 154 amino acid residues [38]. Hydrogen atoms were added to the protein. The histidine (His) residues were considered to be in a neutral state (HSE). Among these five samples, the E3 binding domain of the dihydrothiamide in the 2-oxyglutarate dehydrogenase (2-OGDH) complex of *Escherichia coli* (PDB code: 1BBL) was investigated as a typical sample in the present work. The mutations that are in the residues at an active site or any other residue at the distant site affect the structure of proteins, substrate, and cofactor binding site. Mutations do not just provide us an understanding of novel protein design with enhanced stability and activity, but it also gives us a platform where we can design mutants with a strong understanding of the importance of a particular amino acid at a specific location, having a stability effect. The wild-type protein 1BBL has a net positive charge. The 6 types of mutants of 1BBL were achieved by replacing the charged residues. For example, aspartic acid (Asp) and asparagin (Asn) are often transformed into each other in nature. The mutant 1, protein D43N, was obtained using aspartic acid (Asp) instead of No. 43 residue asparagine (Asn) from the wild-type protein. The side chains of aspartic acid containing carboxyl groups, which give it a negative charge under physiological conditions. The uncharged asparagine has the polar side chain, which can form hydrogen bonds appearing at the corner. Then the mutant D43N has two positive charges. The chloride ions were added to the system to neutralize it. The protein configuration was enclosed in the center of the periodic cubic simulation box. The detailed charged conditions of the 1BBL and its mutants are listed in Table 2.

All simulation procedures were performed by molecular dynamic algorithms implemented in the NAMD 2.6 software package (Beckman Institute, University of Illinois at Urbana-Champaign, Urbana Troop, IL, USA) using an all-atom CHARMM27 force field [39]. First, the simulation system was energy minimized under the convergence criterion of the maximum force value 10 kJ/nm/mol by the steepest descent method for 60,000 steps. Then, 100 ps equilibrations were performed at the constant temperature and pressure ensemble (NPT). The molecular dynamics simulations were carried out for 10 ns under the condition of constant temperature *T* = 310 K and constant pressure *P* = 1 atm. The time step and mesh spacing were set as 2 fs and 0.1 nm, respectively. The van der Waals interactions were calculated by the switching function, which start at a distance of 1.0 nm and reach zero at 1.2 nm. The long-range electrostatic interactions were calculated by the particle grid Ewald (PME) method. The protein configuration was enclosed in the center of a periodic cubic simulation box with TIP3 waters filled in. A multiple time stepping integration scheme was applied by NAMD. Following the equilibration stage, alternating electric fields with different forms and frequencies were applied. The pulsed and oscillatory forms were adopted to express the external electric field. All electric fields were applied along the x-direction with the strength 0.5 V/nm. The detailed parameters of various electric fields are listed in Table 3. The effects of the alternating electric field on its secondary structure of the protein 1BBL and its mutants were estimated by the STRIDE algorithm implemented in the VMD software package [40]. It helps to simplify the analysis of the tertiary conformation of a protein by assigning different types of secondary structure to each residue based on the knowledge-based algorithm, which takes into account the hydrogen bond energy and statistically derives the information on the torsional angles of the protein.

In this study, we carried out MD simulations to explore the effects of alternating electric fields, varying in frequency and electric form, on the protein samples and their mutants. The conformational stability of proteins during the simulation procedure was examined by calculating the root mean square deviation (RMSD) and the root mean square fluctuation (RMSF). RMSD is a numerical measurement of the conformational changes between two structures. It is defined as:(1)$$RMSD=\sqrt{\frac{1}{N}\sum_{i=1}^{N}{\left|r_{final}(i)-r_{initial}(i)\right|}^{2}}$$
where *N* is the number of protein atoms. *r_final_*(*i*) and *r_initial_*(*i*) are the coordinates of an atom *i* in its final structure and initial structure, respectively. RMSF is an important tool to characterize the freedom of the center $C_{\alpha }$ atoms in protein molecules, which is defined as:(2)$$RMSF_{i}=\sqrt{\frac{1}{t_{total}}\sum_{t_{j}=1}^{t_{total}}{\left|r_{i}(t_{j})-r_{i}{}^{ref}\right|}^{2}}$$
where *t*_total_ is the total simulation time, and the reference coordinate *r^ref^* is the average coordinate of $C_{\alpha }$ atom during the whole simulation period. In general, proteins possess electric dipole moment by virtue of their structure constructed by some charged amino acids, such as Lysine, Arginine, Aspartic acid, etc. Once an external electric field is applied on the protein, it induces a realignment charge with respect to the direction of the electric field. The dipole moment is defined as
(3)$$\vec{d}=\sum_{i=1}^{N}q_{i}(i){\vec{r}}_{i}$$
where *N* is the total number of protein atoms. $q_{i}$ is the charge of the atom *i*, ${\vec{r}}_{i}$ is the directional vector of each atom, and the relation $d_{}^{2}=d_{x}^{2}+d_{y}^{2}+d_{z}^{2}$ holds.

## 3. Results and Discussion

### 3.1. Effect of the Frequency of Electric Fields

To investigate the effect of the frequency of the external electric field on the molecular protein, a series of MD simulations were carried out by applying the electric fields with the same electric intensity (0.5 V/nm) but different frequencies (1000/2000 MHz). The time evolutions of three components of a dipole moment during 10 ns simulation are shown in Figure 1. The protein 1BBL was exposed in the pulsed or oscillating types of electric fields with the same maximum electric strength. Figure 1a,b shows that the sign and oscillatory frequency of an *x*-component dipole moment maintain the same directions and frequencies as that of the external electric fields applied on the protein. Figure 1c,d presents the sinusoidal curves when the protein is exposed in the electric field expressing as sine function. In any simulation conditions, the dipole moment along the x-direction significantly increase or decrease with the change of the electric strength, synchronously. The maximum values of the *d_x_* in these four figures are equal in magnitude, which are directly related to the same intensity of the external electric fields. Meanwhile, the *y* and *z* components of the dipole moment show the small fluctuations between positive and negative values. In addition, the average value over the simulation time is close to zero. We concluded that the magnitude of dipole moment had a positive relation with the amplitude of the external electric field [35]. Figure 1 clearly shows that the oscillographs and frequencies of the dipole moment *d_x_* are the same as the external electric fields applied along the *x*-direction. It indicates that the general effect of applied electric fields on the dipole moment is to increase its magnitude along the same direction, simultaneously. Green and Kubo showed in serial papers that transport coefficients, such as heat conductivity and bulk viscosity, can be related to the correlation functions of the corresponding flux or tensor in thermal equilibrium [41,42]. The Green–Kubo formula offers a simple way of relating the diffusive properties of a system and its velocity dynamics. For any physically observable parameter of interest, its thermal average at the perturbed non-equilibrium state can be expressed as the convolution of the external force and generalized susceptibility. When a protein system at thermal equilibrium is slightly perturbed by an external electric force, the response of the system as a function of frequency could, in principle, be predicted from the time correlation function of dipole moment fluctuations at the equilibrium state by the Green–Kubo relation. It will be investigated in the near future.

The time evolutions of the RMSD of 1BBL exposed in electric fields with the same intensities but different frequencies and forms are shown in Figure 2. When 1BBL is exposed in the electric fields with the intensity of 0.5 V/nm, RMSDs show the obvious increase comparing the ones in the simulation condition of without an electric field. From these figures, one can notice that the deviations in the RMSD plot indicate the structural disruption from its native conformation. Also, the visualizing fluctuations of RMSD within 2.5 nm over 10 ns simulations present the dynamical changes of the protein sample with the rapid shifts of electric fields. Figure 2a,b also clearly shows that the values of RMSD change more quickly in the condition of the external electric field with a high frequency. Meanwhile, most values of RMSD are much lower when the protein exposed in the electric fields with a frequency of 2000 MHz than the ones in the electric fields with a frequency of 1000 MHz, except for the initial stage within 1.5 ns. It means that the frequency of the electric field influences the rapidity of the conformational change. In addition, the lower frequency of an alternating electric field makes for a lower structural stability of the protein. Rinne et al. performed extensive molecular dynamics simulations on a single eight-residue alanine polypeptide in explicit water to investigate the influence of α-helix formation on the dielectric spectrum [43]. They found that for such a short polypeptide, the maximum adsorption was centered around a frequency of 250 MHz. According to the scaling theory that gives the connection between the frequency and rotational relaxation time of the protein, the protein needs longer rotational relaxation time when the characteristic frequencies are even lower. It could provide the reason that the protein sample 1BBL presents higher RMSDs when it was exposed to the external electric field with a lower frequency.

To investigate the general effect of the frequency on the all-α protein samples with different α-helix ratios, the average values of RMSD during the whole simulation procedure are presented in Figure 3. As the same results obtained from Figure 2, for certain protein sample, the value of RMSD of the protein exposed in the altering electric fields with high frequency is less than the one in the case of low frequency conditions. It indicates that the protein presents a more stable conformation. Another attractive tendency is that the average value of RMSD decreases with the increase of the α-helix ratio, especially when the protein is subjected to oscillating electric fields. The protein with more residues participating in forming α-helices, which as the stable conformation on the secondary structural level, can present stronger stability even with the influence of electric field. Among these five protein samples, 1BBL presents the smallest difference between alternating electric fields with different frequencies and curve forms. The following discussions are mainly about 1BBL, which has relatively coincident structural stability but still shows some typical characteristics under different electric fields.

The effect of the frequency of the electric field on the secondary structure of the protein 1BBL was estimated by the VMD software package. Figure 4a,b presents the stride evolutions of secondary structures of the sample 1BBL exposed in the oscillating electric fields with frequencies of 1000 MHz and 2000 MHz, respectively. The residues from No. 14 to 23 (labeled as segment I) and No. 41 to 47 (labeled as segment II) comprise two alpha-helix segments in its initial conformation. As seen in the figures, the segments I and II both transformed from helical regions (assigned as the color purple) to turns (assigned as the color cyan) or random coils (assigned as the color white) under the condition of the oscillating electric field. The conformational change in segment II is earlier and more drastic than that in segment I during the simulation process. Comparing Figure 4a,b, the simulation results also show that the secondary structure transition of the protein 1BBL exposed in the electric field with a frequency of 2000 MHz occurs in a much earlier stage than when it is exposed to the electric field with a frequency of 1000 MHz. As shown in Figure 2b, the obvious increase of RMSD was obtained much earlier when the higher frequency was applied on the protein sample. It means that the quick changes of the electric field induce the quick transitions of the secondary structure of the protein. The rapid despiralization of the helix structure into either turns or random coils demonstrates the violent conformational destabilization of the protein when it is subjected to an electric field with a high frequency.

### 3.2. Effect of the Type of Electric Fields

The time evolutions of the RMSD of 1BBL exposed in different types of electric fields with the same intensities and frequencies were presented by comparing Figure 2a,b. As seen in these figures, the value of RMSDs fluctuates without an obvious plateau. This indicates that the protein subjected to the alternating electric fields was stretched or compressed along the x-direction periodically. Figure 2a,b also shows that the RMSD of the protein exposed in the oscillating electric fields are slightly lower than those under the pulsed condition when the frequency of the electric fields are the same. In addition, Figure 3 shows a more obvious tendency that the average RMSDs of the other four protein samples exposed in the oscillating electric fields is lower than the ones exposed in the pulsed electric field. The effect of the type of electric fields is more significant than the frequency on the structural stability of the protein. The reason is that the average strength during the complete period of the oscillating electric field is smaller than that of the pulsed electric field. According to our previous work, the structural stability is strongly related to the electric strength [35].

Besides the discussion of RMSD, RMSF also gives us some useful information to know the structural properties of the molecular chain. The RMSFs of skeleton $C_{\alpha }$ atoms in 1BBL subjected to different types of electric fields are shown in Figure 5. The curves of RMSF show that the two ends of the protein chain have higher movement than the middle part of the molecular protein. It is easily concluded that the end parts of the protein show various structural diversity since they have high freedom during MD simulations. The middle part of the protein has the lower degree of freedom because of the relatively confined space occupied by nearby atoms. Under the electric fields with certain intensity and frequency conditions, the average value of the RMSF of the whole protein sample subjected to the oscillating electric fields is slightly lower than that in the pulsed type of electric field. It is the same trend as the results concluded in Figure 2 and Figure 3.

The stride evolutions of secondary structures of 1BBL in the different types of electric fields with the frequency 1000 MHz are shown in Figure 6. Although, a small gap exists in the numerical difference between the average values of RMSF when 1BBL is subjected to the different types of alternating electric fields. The secondary structure transitions occur clearly in both segments I and II. Figure 6a shows that the structural transitions of segment I and II happen almost simultaneously when the protein sample is exposed in the pulsed electric field. However, when the oscillating electric field is applied on the protein 1BBL, these two fragments present different conformational stabilities in Figure 6b. The secondary structure transition of segment II was much earlier than what occurs in segment I when the protein is exposed in the oscillatory type of electric field. So, these two figures indicate the effects of different types of electric field on the secondary structures’ transition. More rapidly and drastically structural transitions appeared when 1BBL was exposed in the pulsed electric fields than in the oscillating electric fields. Compared with the results concluded from RMSDs or RMSFs, secondary structure analysis is significant to distinguish the effect of the type of electric fields on the protein.

### 3.3. Effect of the Mutations

In order to investigate the relationship between the quantity of electric charge among the molecular protein and its conformational properties, the wild-type 1BBL and its mutants in the external electric field were examined by MD simulations. The time evolutions of RMSD of 1BBL and D43N exposed in the electric fields with different frequencies and types are shown in Figure 7. These four figures show that the values of RMSD fluctuate more obviously in the case of wild-type 1BBL than that of D43N when both of them are exposed in the same electric condition. The reason is that the wild-type protein has one more negatively charged residue than D43N. This may reflect that the charged residues receive the effect of an electric field by perturbing peptide conformation. The electric charges in the protein lead to the involvement of the localized dipolar alignment. Meanwhile, Figure 7a,b presents a more evident difference of RMSDs between 1BBL and D43N than the results shown in Figure 7c,d. According to the simulation results in Figure 2, we conclude that 1BBL in the electric field with the lower frequency presents the lower structural stability. It also demonstrates that the electric charges among the molecular protein have the high correlation with its conformational properties when the protein is in the electric field with relatively low frequency.

The stride evolutions of secondary structures of 1BBL and D43N under the oscillating electric field with a frequency of 2000 MHz are shown in Figure 8a,b, respectively. As the same results obtained from Figure 4, two alpha-helix segments present the different characteristic times when the structural transition occurs. As seen in the graphs, the helical structure of Segment I was maintained for a longer time than Segment II. Segment I showed the higher conformational stability than that of Segment II when the oscillating type of electric field was applied on the protein samples. Comparing secondary structural analysis on 1BBL and D43N, the transitions of segments I and II in 1BBL are both earlier than that in D43N. Although the residue sequences of segment I in 1BBL and D43N are the same, it also presents different conformational stability. D43N, which has one charged residue less than the one of 1BBL, presents the relatively stable conformation during the MD simulation process. The changes of the secondary structure in the wild-type protein 1BBL are faster and more dramatic than that of its mutant D43N, indicating the great influence by the external electric field on the charged residues and even the whole molecular conformation as well.

In order to further explore the relationship between the charge distribution and conformational properties of protein molecules, D43N and other mutants of 1BBL obtained by changing several residues were investigated. The average values of RMSD of 1BBL and six mutants exposed in the same external condition OEF2 are presented in Figure 9. Mutant 1, 2, and 3 were obtained by replacing the negatively charged residues to uncharged ones. Along the black curve with symbols, the net charge of the total protein samples increases from wild-type 1BBL to Mutant 3, showing that the larger value of the average RMSD achieved from the protein consists of more negatively charged residues. On the contrary, Mutant 4, 5, and 6 were obtained by replacing the uncharged residues to positive ones. Although the net charge among Mutant 1 and 4, Mutant 3 and 5, and Mutant 4 and 6 are the same, they present the different value of average RMSDs. If the protein has more charged residues, its structural stability decreases as the increase of average RMSD. The value of average RMSD is independent on the net charge of the protein. This indicates that the direct effect of applied electric fields on the charged residues influences the whole conformation of the protein.

## 4. Conclusions

In this work, the effects of external electric fields with different frequencies and electric types on the wild-type protein 1BBL and its mutants were investigated by MD simulations. The conformational stability and the structural transition of the protein samples were discussed according to our simulation results. The dipole moment analysis showed that the external electric field could affect the conformation of the protein by changing its axial component of the dipole moment. The magnitude and frequency of the dipole moment is directly related to the strength and frequency of the external electric field. The curve oscillographs and frequencies of the dipole moment are the same as the external electric fields applied on the protein. It also concluded that the frequency of the electric field influenced the rapid response of the conformational change. Meanwhile, the protein presented the low conformational stability when it was subjected to the electric field with a low frequency. The protein consisting of more α-helices presents higher conformational stability even when exposed in the electric fields. In terms of the type of electric fields, the average values of RMSD and RMSF of a whole molecular protein are slightly larger in the pulsed electric field which has the higher average intensity over the simulation time than the oscillating electric field. According to the analysis of the secondary structure, two alpha-helix segments transit to turns or random coils almost simultaneously when the protein was exposed in the pulsed electric field. However, two segments present different characteristic times when the transition occurs in the condition of an oscillating electric field. Moreover, the influences of the quantity of electric charge among the molecular protein on its conformational stability were analyzed by the RMSD trajectory of wild-type protein 1BBL and its mutant, D43N, exposed in the pulsed and oscillating electric fields. The net charge dependence of average RMSD of 1BBL and its six mutants showed that the charged residues receive the effect of an electric field by perturbing peptide conformation. The protein with more charged residues had the low conformational stability. The conclusions derived from MD simulation will be helpful to understand the effect of the frequency and expression form of external electric fields on the conformational changes of the protein with charged residues. These findings might potentially impact and provide insights into the mechanism of biological effects of nanosecond-scale intense electric pulses and oscillating electric fields, which are being experimentally explored for their biomedical applications.

## Figures and Tables

**Figure 1 polymers-14-00123-f001:**
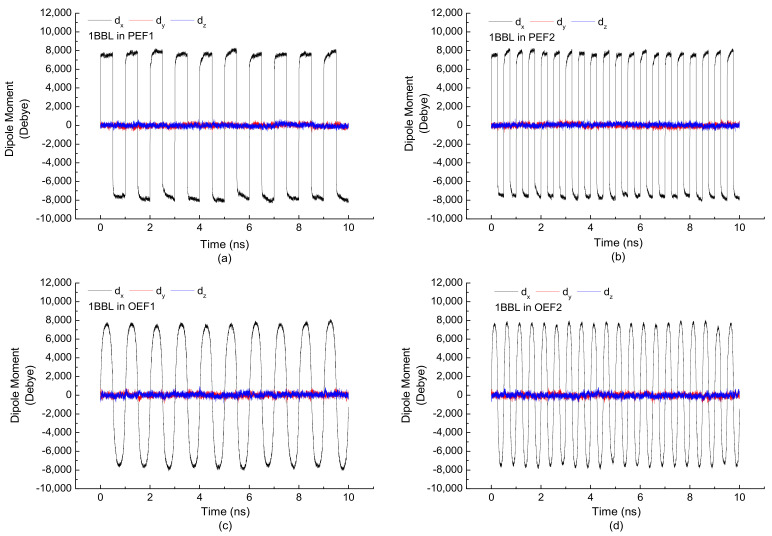
(Color online) Time evolution of three components of the dipole moment when the protein 1BBL is exposed in the (**a**) pulsed electric field with a frequency of 1000 MHz, (**b**) pulsed electric field with a frequency of 2000 MHz, (**c**) oscillating electric field with a frequency of 1000 MHz, and (**d**) oscillating electric field with a frequency of 2000 MHz, respectively.

**Figure 2 polymers-14-00123-f002:**
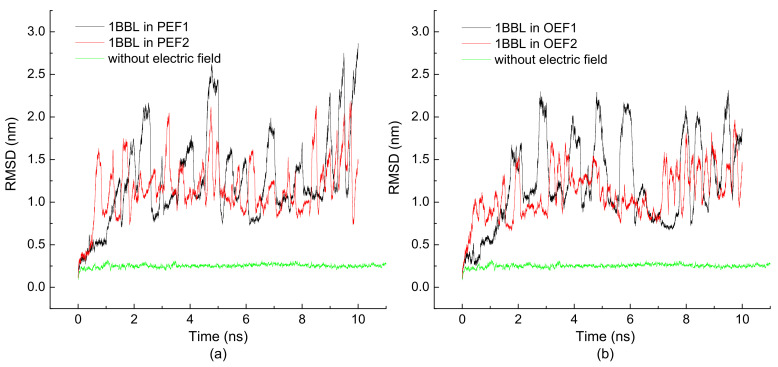
(Color online) Time evolution of the root mean square deviation (RMSD) of the protein 1BBL exposed in the (**a**) pulsed and (**b**) oscillating electric fields with a frequency of 1000/2000 MHz, respectively.

**Figure 3 polymers-14-00123-f003:**
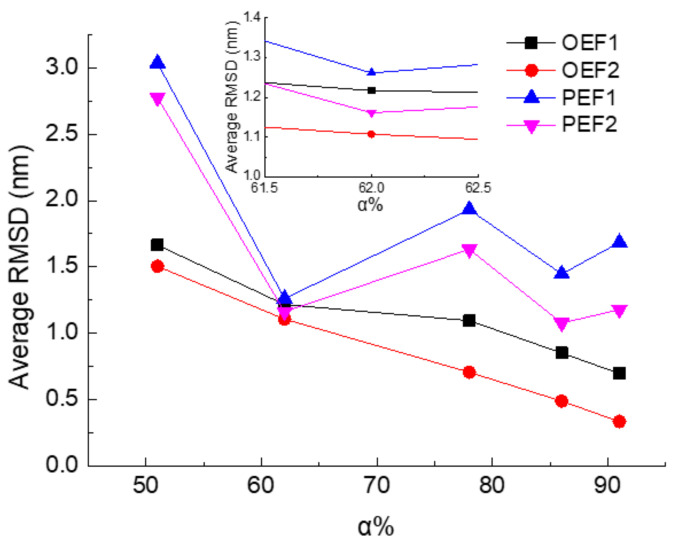
(Color online) The average value of RMSD versus the residual ratio of α-helix among protein samples.

**Figure 4 polymers-14-00123-f004:**
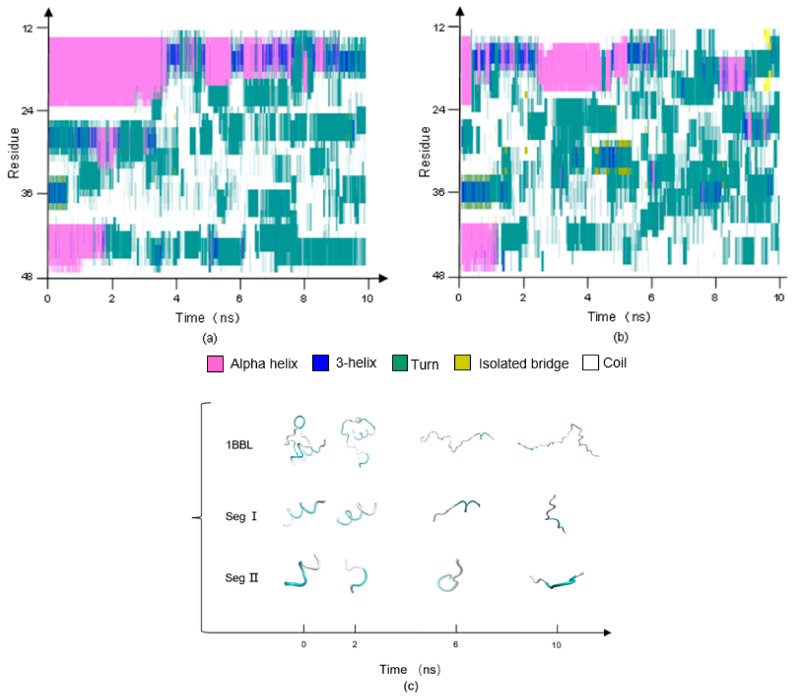
(Color online) Stride evolution of secondary structures of the protein 1BBL exposed in the oscillating electric fields with a frequency of (**a**) 1000 MHz, (**b**) 2000 MHz and (**c**) typical conformation of 1BBL and its segments during the simulation process in the condition of OEF2.

**Figure 5 polymers-14-00123-f005:**
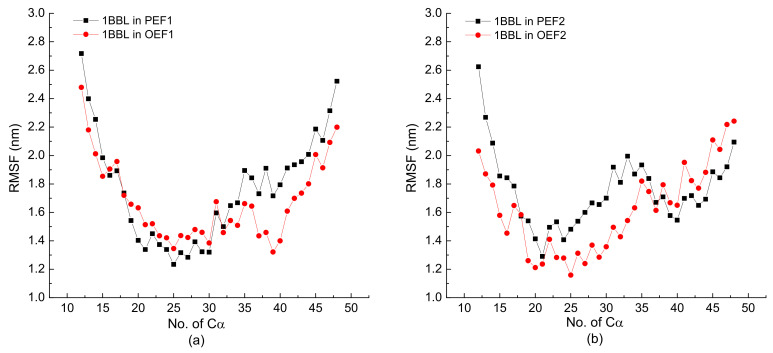
(Color online) The root mean square fluctuation (RMSF) of skeleton $C_{\alpha }$ atoms in the protein 1BBL exposed in the pulsed or oscillating electric fields with a frequency of (**a**) 1000 MHz and (**b**) 2000 MHz, respectively.

**Figure 6 polymers-14-00123-f006:**
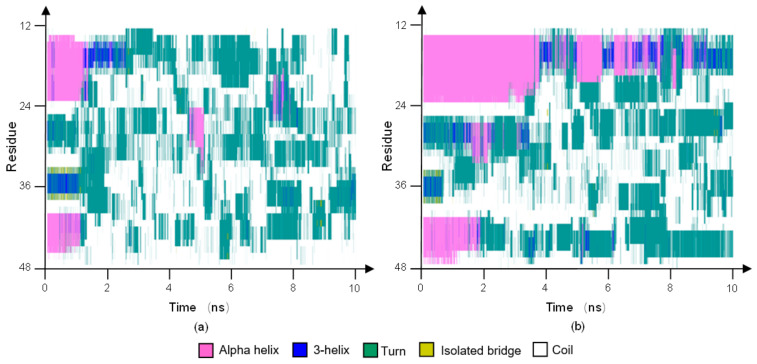
(Color online) Stride evolution of secondary structures of the protein 1BBL exposed in the (**a**) pulsed and (**b**) oscillating electric fields with a frequency of 1000 MHz, respectively.

**Figure 7 polymers-14-00123-f007:**
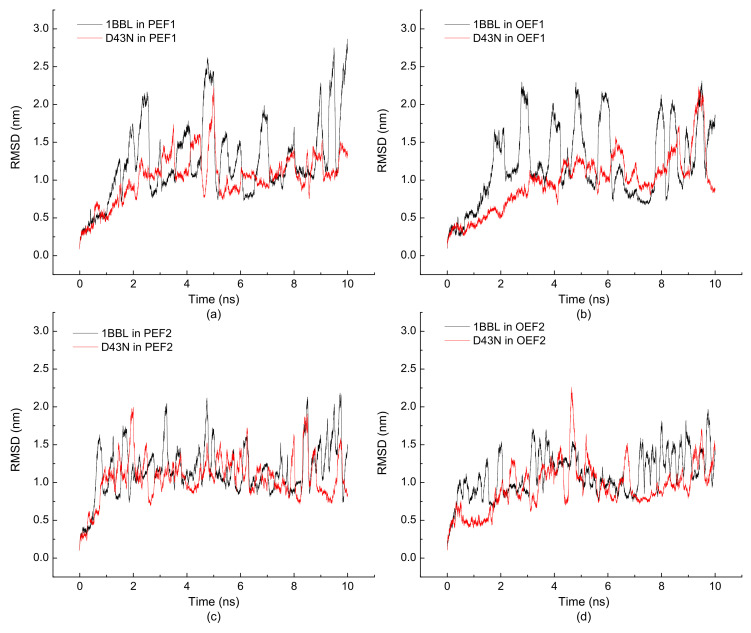
(Color online) Time evolution of the root mean square deviation (RMSD) of the protein 1BBL and its mutant D43N exposed in the (**a**) pulsed electric field with a frequency of 1000 MHz, (**b**) oscillating electric field with a frequency of 1000 MHz, (**c**) pulsed electric field with a frequency of 2000 MHz and (**d**) oscillating electric field with a frequency of 2000 MHz, respectively.

**Figure 8 polymers-14-00123-f008:**
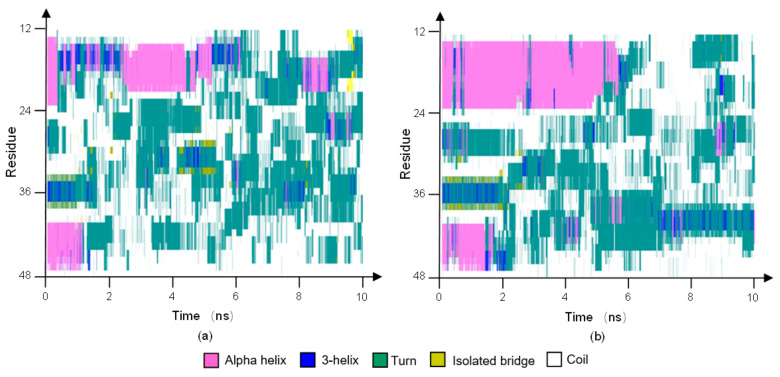
(Color online) Stride evolution of secondary structures of the protein samples (**a**) 1BBL and (**b**) D43N exposed in the oscillating electric field with a frequency of 2000 MHz, respectively.

**Figure 9 polymers-14-00123-f009:**
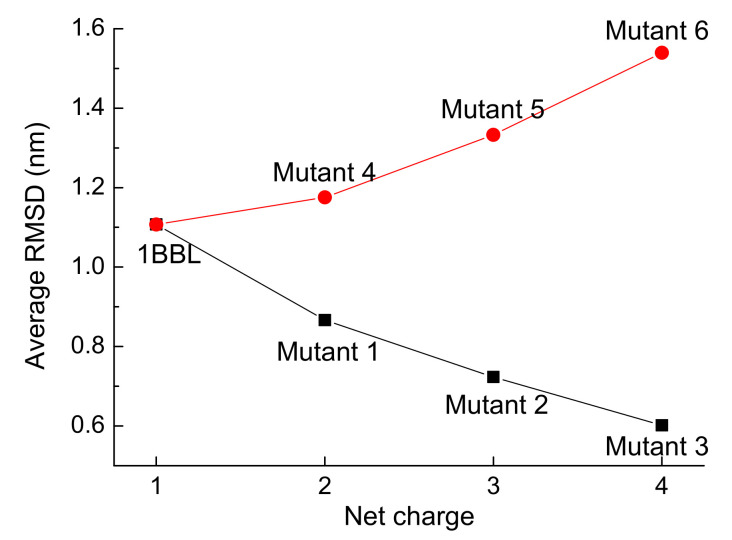
(Color online) The average value of RMSD versus the net charge of the protein samples.

**Table 1 polymers-14-00123-t001:** Structural information of all-α protein samples.

PDB Code	Chain Length	α	β	Others	α%
5CYT	103	52	0	51	50%
1BBL	37	23	0	14	62%
1FCS	154	121	0	33	78%
1F63	154	133	0	21	86%
1EA8	140	127	0	13	91%

**Table 2 polymers-14-00123-t002:** Charged details of wild-type protein and mutants.

Protein	Wild-Type (1BBL)	Mutant 1 (D43N)	Mutant 2	Mutant 3	Mutant 4	Mutant 5	Mutant 6
Charged residues	11	10	9	8	12	13	14
Negative charge (e)	5	4	3	2	5	5	5
Positive charge (e)	6	6	6	6	7	8	9
Net charge (e)	+1	+2	+3	+4	+2	+3	+4

**Table 3 polymers-14-00123-t003:** Parameters of electric fields.

Electric Field	Type	Frequency (MHz)	Intensity (V/nm)
PEF1	Pulsed Electric Field	1000	0.5
PEF2	Pulsed Electric Field	2000	0.5
OEF1	Oscillating Electric Field	1000	0.5
OEF2	Oscillating Electric Field	2000	0.5

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
