# Peer review of "Effects of an Electric Field on the Conformational Transition of the Protein: Pulsed and Oscillating Electric Fields with Different Frequencies"

_polymers, 2021, doi:10.3390/polym14010123_

Round 1
Reviewer 1 Report
This M.S. is is technically sound,but there are major concerns:
1. There are any bio-related in vitro information of the mutations (table 2) ? It's unclear why the authors used these mutations ?
2. It's unclear that the results can be applied in which field ? medicine ?....
Reviewer 2 Report
This paper reports MD simulation studies on the response of different proteins with high alpha-helix content to an externally applied AC electric field. The authors used two different frequencies (1 GHz and 2 GHz), two different wave shapes and also study different mutations that change the charge of the proteins. The subject is interesting, the simulations are done carefully (as far as I can tell), the interpretation however could be improved.
1) The authors report the dipole moment as a function of time. That the frequency of the dipole moment is the same as the imposed electric field follows directly from response theory, this should be mentioned. The amount of non-linearity can be estimated from the resultant dipole response by extracting the amplitude of higher harmonics, I would suggest the authors do that analysis, it is really straightforward.
2) The response spectrum as a function of frequency could in principle be extracted from the equilibrium dipole fluctuation using a Green Kubo relation. It would be interesting to such a analysis, maybe in the future, at least this should be mentioned in the paper.
3) The absorption spectrum of an alpha-helix forming polypeptide has been calculated in "Impact of secondary structure and hydration water on the dielectric spectrum of poly-alanine and possible relation to the debate on slaved versus slaving water" Rinne et al., THE JOURNAL OF CHEMICAL PHYSICS 142, 215104 (2015). There it has been found that for the relatively short polypeptide the maximum of adsorption was centered around a frequency of 250 MHz and a scaling theory was developed that connects this frequency with the rotational relaxation time of the protein. The proteins the authors study are larger, so rotational relaxation times are longer and the characteristic frequencies even lower. This presumably explains why the protein response at 1 GHz is higher than at 2 GHz, a discussion of this would be a useful addition to the paper.
4) Related to the mechanism behind the observed effects: The authors could split the measured dipole moment into the contribution from the backbone and side chain charges. Since the backbone partial charges produce a sizable dipole moment along an alpha helix, this decomposition could reveal what the effect of alpha-helical dipole orientation is.
5) Likewise, it would be good to analyze how much of the polarization effect is due to rotation and how much due to stretching of the protein.
6) I could not deduce from the manuscript whether the simulations are done with explicit water or not. If the water is not included explicitly, are hydrodynamic effects included? Also what type of time integrator was used?
Round 2
Reviewer 1 Report
The authors have addressed my comments
Reviewer 2 Report
The authors have responded to all comments by the referees and I find the paper ready for publication.
This manuscript is a resubmission of an earlier submission. The following is a list of the peer review reports and author responses from that submission.
Round 1
Reviewer 1 Report
This M.S. interests readers. However, the discussion is weaker. The authors can provide more applications and information about the electric Field on the conformational transition of the proteins.
Reviewer 2 Report
The paper by Qun Zhang and coauthors describes MD simulations of a small protein domain (PDB ID 1bbl) and one point mutant of this protein. Briefly, it comprises 8 short simulations with small systems with very poor data analysis and statistics. The conclusions are not supported by the data (see the details below).
First, 1bbl is not a dihydrolipoamide dehydrogenase, it is only a small domain of this protein. I suppose that it a classification of a small double-helix domain as an all-alpha protein is incorrect.
Selection of 1bbl and the mutation site needs explanation. I wonder why Authors didn't chose larger proteins (more relevant for biology) or (better) a set of proteins with different charge, number of charged residues, size etc.
Statistics is required: at least few simulations with a different protein rotation (or a field direction) should be performed. Any conclusions from single simulation with one field direction are meaningless.
In addition, speculations about the role of charged residues in the effect of the electric field are not supported by data. No structural residue-specific insights were presented. Analysis of the one point mutant is not enough.
Fig.2 - any speculations from noisy curves in such short simulations are meaningless. I don't see significant difference between curves. RMSD changes also should be compared to RMSD of the control: a simple MD simulation without electric field will also give a non-zero RMSD changes.
The conclusions from RMSD graphs looks like over-speculations about the noise on the curves.
RMSF analysis: to claim that the difference is significant/reliable, first check that the protein structure is stable in solution. It is not trivial for a small domain. I wonder that different protein regions showed lower RMSD values in OEF as compared to PEF (fig.5, left/right).
The aforementioned points are true for the section 3.3.
The realism of the particular field parameters should be justified. The fact that Authors observed structural difference in few nanoseconds of the simulations indicates that conditions are too hard and artificial.
With no figures of the typical conformations, it is unclear is there any actual changes and are the changes adequate. Is the protein stable?
Explanation of a standard paper organization (lines 88-92) should be removed.
Figures 2 and 4 present the same curves. Fig.4 should be removed.